# A Community-Engaged Approach to Community Health Needs and Assets Assessment for Public Health Research

**DOI:** 10.3390/ijerph22071030

**Published:** 2025-06-27

**Authors:** Rosanna H. Barrett, Emma Joyce Bicego, Thomas C. Cotton, Supriya Kegley, Kent Key, Charity Starr Mitchell, Kourtnii Farley, Zahra Shahin, LaShawn Hoffman, Dubem Okoye, Kayla Washington, Shawn Walton, Ruben Burney, America Gruner, Terry Ross, Howard W. Grant, Mark V. Mooney, Lawrence A. Sanford, Tabia Henry Akintobi

**Affiliations:** 1Department of Community Health and Preventive Medicine, Prevention Research Center, Morehouse School of Medicine, 720 Westview Drive, Atlanta, GA 30310, USA; robarrett@msm.edu (R.H.B.); chamitchell@msm.edu (C.S.M.); kfarley@msm.edu (K.F.); zshahin@msm.edu (Z.S.); dookoye@gmail.com (D.O.); lsanford@msm.edu (L.A.S.); takintobi@msm.edu (T.H.A.); 2DME Committee, Community Coalition Board, Morehouse School of Medicine Prevention Research Center, 1513 East Cleveland Ave., East Point, GA 30344, USA; tcotton@redeem-advance.org (T.C.C.III); lashawn@lashawnhoffman.com (L.H.); shawn.deangelo.walton@gmail.com (S.W.); ruben_burney@us.aflac.com (R.B.); clila@clila.org (A.G.); lhc55atl@gmail.com (T.R.); hgrant@facaa.org (H.W.G.);; 3Charles Stewart Mott Department of Public Health, College of Human Medicine, Michigan State University, Flint, MI 48502, USA; keykent@msu.edu; 4Center for Program Evaluation and Quality Improvement (PEQI), Emory Centers for Public Health Training and Technical Assistance Operations, Rollins School of Public Health, Emory University, 201 Dowman Drive, Atlanta, GA 30322, USA; kayla.d.washington96@gmail.com

**Keywords:** needs assessment, community-based participatory research, capacity building, co-learning

## Abstract

The Morehouse School of Medicine Prevention Research Center (MSM-PRC) conducted a Community Health Needs and Assets Assessment (CHNAA) survey using a Community-Based Participatory Research (CBPR) approach. In this article, we will demonstrate the application of CBPR in informing research agenda and implementation strategies. We will discuss the practical considerations and potential benefits of engaging the community in data collection, interpretation, and utilization to address community health challenges. Emphasizing collaboration, co-learning, and respect, and guided by the CBPR principles, CHNAA ensured that community voices led to the identification and integration of the research priorities. Overseen by the Community Coalition Board (CCB) and its Data Monitoring and Evaluation (DME) Committee, the survey featured closed- and open-ended questions addressing social determinants of health. Out of 1000 targeted participants, 754 provided valid responses, with a 75% response rate. Most respondents were female and represented a racially diverse group. Descriptive statistics and thematic analysis revealed that key health concerns were diabetes, COVID-19, mental health, and high blood pressure. Barriers to care included lack of food access, affordable housing, and limited mental health services. The findings led to five public health initiatives launched between 2023 and 2024 demonstrating the CBPR model’s effectiveness in aligning community needs with actionable solutions.

## 1. Introduction

Historically, public health priorities and programming development by state and local health department were informed by data collected in disease registries and health surveillance databases designed around specific well-defined objectives. Public health surveillance systems are used to assess public health status, track conditions, define health priorities, evaluate programs, and develop public health research [1]. Health surveillance systems are usually population-based and often lack direct connections with the community beyond the point of data collection or reporting of findings. When communities are not meaningfully involved in the conceptualization, implementation, evaluation, and dissemination of efforts to identify and address local issues, it can erode trust, reduce the cultural relevance of the methodology, and ultimately limit the effectiveness and sustainability of resulting interventions [2,3].

Community-Engaged Research (CEnR) and Community-Based Participatory Research (CBPR) address these limitations by fostering collaborative partnerships in which community members actively participate in data analysis, interpretation, and dissemination of the data that involve their community [4,5].

CEnR, a collaborative approach that actively involves community members, organizations, and other stakeholders in the research process. It is grounded in mutual respect, shared decision-making, and a commitment to addressing community-identified needs to produce meaningful, actionable outcomes [6,7,8]. CEnR values the expertise of both researchers and community members, fostering co-learning and ensuring research remains relevant, ethical, and beneficial [9,10,11]. CEnR is characterized by meaningful involvement of communities in research affecting their well-being, by strengthening trust and collaboration between academic institutions and the community. It provides a framework for working with groups connected by geography, shared interests, or similar experiences to address issues impacting their well-being. CEnR also recognizes the strengths of communities, institutions, and individuals, reinforcing a more inclusive and effective research process [9].

The Continuum of Community-engaged Research (Continuum of CEnR; Figure 1), depicts a variation of different levels of engagement between the community and researchers [9]. In addition, the positionality on the continuum denotes various power distributions between community and institutional partners. The Continuum of CEnR highlights two essential domains to consider, equity indicators and contextual factors. This continuum underscores the importance of these considerations, as they may impact the overall partnership and quality of the research study. These indicators and factors are essential to developing, maintaining or repairing trust which is dependent upon multiple factors that have negatively impacted or positively influenced the researcher’s perceptions of marginalized communities.

One of the widely recognized stages of CEnR is community-based participatory research (CBPR). CBPR is the sixth stage of interaction on the Continuum of CEnR, as depicted in Figure 1 [9], where all the equity indicators and contextual factors are more likely to be fully demonstrated. CBPR has become more widely used in academia to ensure that evidence-based practices are applicable in real-world settings, the community. Most importantly, the CBPR approach highlights the importance of community engagement in research aimed at addressing health disparities and inequities [10,11] through partnership with stakeholders, working in communities of focus. At the intersection of research and community engagement, CBPR presents an invaluable opportunity to create significant and sustainable change in the health of communities [12]. The inclusion of the community in the research process amplifies the existing power of community residents and leaders through pathways of self-efficacy and efforts recognizing the value of their wisdom and the necessity of their leadership through research governance and co-creation [10,11,13,14,15,16,17,18].

The Prevention Research Center (PRC) is the designated center for CBPR within the Morehouse School of Medicine (MSM). The MSM-PRC, funded by the Centers for Disease Control and Prevention since 1998, is focused on gaining knowledge about the best methodologies for solving the nation’s obstinate health problems. The MSM-PRC is nationally recognized for its effectiveness in engaging with local communities in Georgia to co-lead research and public health initiatives. The MSM-PRC has embraced CBPR principles,(Figure 2), as a foundational framework to advance its vision of leading efforts in health equity and justice [12,19].

Central to its mission, the MSM-PRC is dedicated to building collaborative and long-standing partnerships with local communities, demonstrated with the formation and ongoing engagement of a Community Coalition Board (CCB). The CCB was established in 1999 to guide and govern the MSM-PRC’s work. Unlike traditional advisory boards, the CCB operates as a governing and policymaking body, ensuring that community members play a leadership role in shaping research priorities [19]. The CCB is composed of 13 community residents, 3 representatives from academic institutions, and 8 members from health and social service agencies. To reinforce the CBPR principle of recognizing the community as a unit of identity, community members make up more than 50% of the CCB, ensuring that their voices and perspectives remain central to the research process. MSM-PRC staff members provide technical assistance and support to the board. The CCB holds a leadership role in developing, implementing, evaluating, and disseminating research priorities that directly address health disparities [19]. As the MSM-PRC continues its work to bridge gaps in health inequities, the CCB leads the decision-making process for community-driven research initiatives [19,20,21]. The board meets every other month to discuss pressing health challenges, strategize solutions, and advance initiatives aimed at improving community health. This governance model reflects three primary equity indicators of the Continuum of CEnR including established community power and control, decision-making, and influence in the research infrastructure and related process [7]

### 1.1. The Utility of a Community Health Needs (And Assets) Assessment

Community Health Needs Assessments (CHNAs) are traditionally conducted to identify the health needs of the community on specific health topics by capturing community experiences and perceptions. Commonly, CHNAs serve multiple critical purposes in improving public health and have the potential for making significant and sustainable changes in community health [22]. The primary goal of CHNA is to identify health needs by understanding the physical, mental, and social well-being challenges faced by a community [22,23] but studies have found that CHNAs do not always engage the community in the analysis, interpretation, and explanation of the findings [23,24,25]. In order to prevent portraying the community in a deficit manner, it is important to also examine the local health assets that have kept the community resilient through various health challenges for decades. Conducting a Community Health Needs and Assets Assessment (CHNAA) addresses this aspect. In general, CHNAA should be conducted to examine community concerns as well as local resources and strengths that can be leveraged to address these health concerns [16,26]. A key aspect of the process is community engagement, ensuring that stakeholders and community residents equitably collaborate, fostering a *sense of ownership and collaboration* in the processes and outcomes of the assessment—a CBPR principle. CHNAA can play a vital role in identifying and addressing disparities in health outcomes and access to care, especially among vulnerable and marginalized populations. It has shown to optimize resource allocation by directing funding and support to initiatives with the greatest potential for impact [27]. Additionally, by establishing a baseline for monitoring and evaluation, CHNAA enables ongoing assessment of health programs and interventions, allowing for necessary adjustments over time. Together, these objectives make CHNAA a crucial tool for building healthier and more equitable communities [23,24,25].

### 1.2. CBPR-Focused Community Health Needs and Assets Assessment

The MSM-PRC, in collaboration with its CCB, has implemented a CBPR-focused CHNAA process every three to five years to develop a community-centered research agenda. The goal is to better understand and respond to the health needs, assets, experiences, and preferences of the community. As the Center’s reach and relevance grew, partly due to its response, service, and partnerships during the COVID-19 pandemic, an expanded administration of the CHNAA and research agenda was necessary. Resultantly, the MSM-PRC launched the 2022 CHNAA in Georgia’s 40 rural and 5 urban counties. The CHNAA serves as a CBPR mechanism by which research questions and ideas are drawn from community’s expressed health concerns and preferences through a collaborative process.

This article details CHNAA processes and the application of CBPR principles along with Continuum of CEnR factors and indicators guiding implementation and reflected in outcomes. It will also illustrate how to inform public health research that stimulates research participation. The article also demonstrates how community and researchers can work together to secure funding and implement research that addresses the needs, priorities, and strengths of communities. Meaningful community engagement is critical to protect its values, needs, and concerns.

## 2. Materials and Methods

A descriptive study design was applied to assess the influence of the community in research planning and implementation. In CBPR, community input is valued and respected. Therefore, all CHNAA activities, from survey development to data analysis, interpretation, and application, were reviewed, approved, and monitored by the CCB led by a subgroup of the CCB, known as the Data Monitoring and Evaluation Committee (DME). The DME was established by the MSM-PRC in 2009 to extend and enhance community engagement of CCB members in data and evaluation processes and to extend community oversight and direction of the work of the MSM-PRC. The DME comprised eight members that existed through the co-leadership of both a CCB community member and the MSM-PRC Evaluation Director. This community oversight and co-leadership embodies the CBPR principle that facilitates *a collaborative and equitable partnership in all phases of research*.

The DME was responsible for reviewing, preparing, and ensuring that CHNAA results are presented to the full CCB body. Scheduled bi-monthly and ad hoc meetings were held to facilitate collaborative engagement with community members respectfully and transparently and to *foster co-learning and capacity building*, a principle of CBPR. DME members meet every other month, alternating with the larger CCB meeting, to discuss and inform the CHNAA survey and administration implementation plans. The CCB received bi-monthly progress reports related to CHNAA progress, and the members provided valuable feedback on how best to address identified challenges. The CCB reviewed the recommendations of the DME during general bi-monthly meetings and provided input which was critical for deciding on the final documents of CHNAA before distribution to the community. Recommendations for addressing these challenges were expediently acted upon to ensure a smooth and seamless process.

### 2.1. CHNAA Survey Instrument Development

The 2022 CHNAA survey, Appendix A, was designed to apply the CBPR principle that *focuses on identifying locally relevant public health problems and the ecological perspectives that attend to the multiple determinants of health*. The open-ended questions were geared towards capturing community perspectives and lived experiences associated with health concerns as well as social, environmental, and policy-related factors that need to be addressed to improve health outcomes. Quantitative questions focused on defining the population surveyed to determine if all voices of those who often portray poor health outcomes were well represented. MSM-PRC collaborated with the DME to revise the survey length, assess comprehension, and assure face validity of the survey. After final approval by the DME, the survey questions, survey guide, and other supporting materials were submitted to and approved by the MSM Institutional Review Board.

### 2.2. CHNAA Administration

The CHNAA is not considered a research study by the MSM Institutional Review Board (IRB). Therefore, the MSM IRB provided an exemption in the first year of the administration of the survey which remains valid for all the consecutive years (2017–2022) it was administered by MSM-PRC. Before collecting information, survey participants were given frequently asked questions (FAQs) detailing the purpose and procedure for the assessment. It was clearly stated that participation is voluntary and optional. Participants were also informed of their rights to discontinue taking the survey at any time. Therefore, by completing the survey, participants indicated their informed consent. In 2022, the CHNAA survey was conducted from April to September online or in person.

Throughout the CHNAA development and administration process, the MSM PRC relied on community input based on trusted relationships built over the 20-plus years of engagement with the CCB. Through this community-campus partnership, the MSM-PRC gleaned support from the CCB to support effective CHNAA promotion, data collection, and interpretation. This approach is detailed in the sections that follow to demonstrate the actualization of the CBPR principle that *builds strengths and resources in the community through shared knowledge, commitment, and collaboration* to address the health needs of the community while *preserving its assets.*

#### 2.2.1. Sampling Methods

In previous years of administration, the CHNAA has only historically been disseminated within the Neighborhood Planning Units* (NPU) throughout the City of Atlanta with sampling focused on Fulton County, Atlanta NPUs, and other urban counties across Georgia.


*The City of Atlanta is divided into twenty-five (25) Neighborhood Planning Units (NPUs), which are citizen advisory councils that make recommendations to the Mayor and City Council on zoning, land use, and other planning-related matters.*

*Source: Neighborhood Planning Units| Atlanta, GA*


In this iteration of the CHNAA, other urban and rural counties were included. To guide this broader inclusion, we utilized the Center for Disease Control and Prevention’s (CDC) Social Vulnerability Index (SVI) to help identify communities most in need of representation in the assessment. SVI refers to the potential negative effects on communities caused by external stressors on human health. Such stressors include natural, or human-caused disasters, or disease outbreaks [28]. Reducing social vulnerability can decrease both human suffering as well as economic loss. The CDC SVI utilizes 16 different U.S. Census variables to help officials identify communities that may need support during or after disasters. In much the same way and aligning with the PRC’s mission to continue to bridge the gap in health inequities, the assessment intentionally decided to sample other counties that were also disadvantaged and/or underserved. The survey had a goal to collect responses from 1000 community members.

#### 2.2.2. CHNAA Data Collection Methods

The CHNAA survey data collection process involved a mixed-methods approach using both closed-ended (quantitative) and open-ended (qualitative) survey questions. This allows for a more in-depth knowledge and awareness of the community’s concerns and challenges. Evidence suggests that utilizing a mixed methods approach does allow for contextualization of findings while adding rich data to conclusions due to the addition of the qualitative piece. Additionally, utilizing different methods to collect data on the same subject can also increase the credibility and rigor of results [29,30]. The DME members assisted in the framing of survey questions to ensure that they were clear and easily understood. The survey questions were administered to capture demographic, social, and economic information and participant thoughts, perceptions, and experiences related to health and healthcare concerns. To reduce the risk of self-reporting bias, the survey questions were clear and concise, and leading questions were avoided.

Recruitment for distribution of the CHNAA survey utilized several strategies that were centered around community reach and accessibility. First, a combination of MSM-PRC staff, CCB members, and other community partners disseminated surveys at community-based events, neighborhood meetings, recreational facilities, libraries, and health clinics. The community members had an option to either take the paper-based survey in-person or as an e-survey through a Qualtrics survey link or a QR code. Additionally, CHNAA recruitment flyers were also shared at community-based events, neighborhood meetings, recreational facilities, libraries, and health clinics to inform community members of the survey. Of great help in the recruitment effort was the MSM Office of Community Engagement (OCE), which aims to promote and strengthen effective partnerships between faculty, staff, and students and those who identify as community residents and leaders, government agencies, faith-based organizations, or health and social service agencies. The OCE staff partnered with the MSM-PRC evaluation team to share the CHNAA recruitment flyers at community partner events to garner community awareness and participation. The OCE Community Health Workers’ (CHWs) have established trust and proximity to the community which enabled the MSM-PRC evaluation team to better navigate survey distribution during community-facing events.

Data was collected online via Qualtrics, a secure web-based survey tool, and in-person at community events. The survey took approximately 25 min for each survey participant to complete. Community members who completed the survey in-person, at community engagement events, received monetary compensation of a $25 physical gift card at the point of contact. After validation of the authenticity of their contact information, individuals who completed the electronic survey were compensated with a $25 electronic gift card sent to them via their authorized email addresses t To protect the identity of the survey participants, their email information was only available to the MSM-PRC research staff and used only to send the e-gift card. The compensation of $25 per survey participant was based on standard rate and was determined by the estimated period for completion of the survey (participation level) to avoid under or overcompensation for time and effort.

Cross-validation checks were performed to remove erroneous results and invalid entries. Bot-generated responses and responses from unexpected locations, including out-of-state, were removed from the online survey data pool. In-person surveys were done with the supervision of team members. Qualtrics XM, the survey software used, also has fraud detection features which helped to eliminate duplicate responses. Survey results were also scanned for illogical answers to open ended questions and inconsistencies in sequential questions. All survey data were stored in a secured password protected portal accessible only to the evaluation team to prevent unauthorized access, modification, or deletion of data.

#### 2.2.3. Data Analysis Methods

Data analysis was conducted by a team of experienced PRC evaluators. The DME members did not participate in actual analysis of the data but rather the display and interpretation of survey findings. The quantitative data of the CHNAA survey was analyzed using the IBM Statistical Package for the Social Sciences (SPSS) software Version 27.0 [31]. The qualitative survey data were analyzed using Dedoose, a web-based qualitative analytical application for research and program evaluation [32]. The qualitative survey responses were coded by four independent coders and analyzed for common themes by using thematic analysis as a means for assessing patterns and meanings that were present throughout participant responses. First, the coders independently familiarized themselves with the data and took note of their first impressions. The generated initial codes describe the overall content of the data, patterns, and themes across the data. Themes were then checked to determine if they were consistent and coherent. Consensus coding was performed when an additional coder was needed to compare coding results. Lastly, the scope and significance of common themes were identified and described. Key quotations from community members were also used to highlight the richness of data in participant responses and to amplify community voices through their spoken words. This information was shared among PRC-affiliated researchers, community members, and organizations to use for their health promotion initiative planning. The CHNAA process flow is depicted in Figure 3.

## 3. Results

### 3.1. CHNAA Quantitative Findings

The CHNAA survey collected data from 754 community residents throughout the state of Georgia (16.5% paper and 83.5% electronic surveys). Many survey respondents did reside in the Fulton County area (57%) while other counties that were represented in the sample population included, but were not limited to Dougherty, Cobb, Gwinnett, and Dekalb. Survey respondents were also largely female (62.2%) and more than 49% of the study population did identify as Black and/or African American. 39% of the survey population identified as White. More than 50% of the respondents were aged between 25–44 with 25.5% aged between 25–34 and 25.6% aged between 35–44 years. By Race/Ethnicity, 49% of respondents were Black/African American, and 40% were White; about 79 percent identified as non-Hispanic, 11 percent as Hispanic, and 84% of individuals identified as heterosexual. Additionally, 53% of individuals indicated that they were married while 28% noted that they were single and never married. 36% of people noted that they lived in a 3-person household. Most respondents’ family income was between $40,001–$55,000 and $55,001–$75,000.

Respondents reported on the top individual health concerns, community concerns, health-related, systems, or policy-based issues (social determinants of health) and why these issues were important to them. Respondents were allowed to select multiple issues. The top five community health concerns identified by respondents were Diabetes (24%, n=183), COVID-19 (23%, n = 176), Mental Health (20%, n = 154), high blood pressure (20%, n = 151), and heart disease (17%, n = 128). When asked about their own health, the ordering of the top five health concerns changed, with COVID-19 (19%, n = 140) and Mental Health (19%, n = 140) superseding Diabetes (18%, n = 137), and followed by high blood pressure (18%, n = 135), and Cancer (17%, n = 126). Mental health treatment was also identified as one of the top three most frequently cited health-related policy, system, and environmental issues. More details are presented in Appendix A. Other information collected through the CHNAA was health care access and health promotion strategies.

### 3.2. CHNAA Qualitative Findings

To amplify the voices of the community, survey respondents were asked why the health concerns they identified were important, what they considered to be the causes of the health issues, and to suggest solutions to address those health concerns. More than 70% of the respondents shared their reasons, and what they considered a solution to the problem. Overall, the top community health concerns were important to respondents because: (1) they were related to their personal or community health; (2) they weren’t knowledgeable about the health issue (s); or (3) they lost their loved one to the health concern (s).

To expand on this information, one respondent expressed


*“They tend to run in my family so being more educated would definitely help better understand or at least know where to seek help.” Another individual cited, “Because these diseases are afflicting me and my family.”*


Respondents thought that the causes for the health concern (s) were a lack of resources and knowledge about the health issue (s), an unhealthy diet, and a lack of exercise. One participant cited

*“Lack of health education, health screening programs…”* Another individual stated
*“Businesses that have better/cleaner options aren’t in the neighborhood. Grocery stores don’t make it a priority to provide fresh produce.”*


Furthermore, participants stated that more resources, health education programs, and access to healthcare services and treatment would solve these health issues. Specifically, one individual expressed


*“Creating health screening and education programs, offering free/low-cost primary health care and incentives (free medicine and supplies) to working people for adhering to treatment and control of these issues.”*


Another respondent expressed *“If these issues are adequately addressed the overall community would improve in these areas.”* Another individual stated, *“Only by expanding the scope of education and increasing the content of education can we effectively avoid the occurrence of health problems.”*

Respondents were asked how community members could be best informed about health programs from the MSM-PRC, and the responses included community events, church, and social media. Participants noted that the top ways that the MSM-PRC can share health information with them were through email, attending community events, and through Facebook.

### 3.3. CCB-Engagement Assessment

CCB members were surveyed to assess Continuum of CEnR contextual factors and equity indicators reflected in the CHNAA process. The members were asked to identify the contextual factor they perceived as most appropriately addressed. Trust and Transparency emerged as the most frequently selected, both chosen by 22% of respondents. Relationship Building and Respect were also prominent, each identified by 20% of participants. History was selected by 14% of respondents. In terms of equity indicators, Mutual Understanding was the most selected factor (21%) Decision Making, Influence, and Responsibility were each selected by 14% of respondents, followed by Ownership and Resource Sharing (12%, respectively), and Power and Control (10%).

## 4. Discussion

The CBPR-focused CHNAA intentionally bridged research to practice and moved beyond collecting and recording data on health disparities to informing interventions that promoted health equity among underserved and marginalized population groups through their co-creation of the assessment, its administration, analysis and subsequent research agenda implementation. Furthermore, this community engagement process was designed to collect data on both individual and community-based health priorities, policy or system-based issues to inform responsive strategies. This process should consistently be informed and led by the community. Acknowledging the community as a collaborative and equitable partner is an asset-based approach applied in the CHNAA process to build on the strength and resources within the community [12,19].

The MSM-PRC CBPR-focused CHNAA process involves a powered community, a community through which our CBPR approach acknowledges their existing power. The MSM PRC is not bestowing power to them. Our partnership acknowledges the power of all partners and together we further “power” each other. The community is equipped over time with transferable skills to lead partners in co-creating or updating a CEnR agenda. This recurring practice reflects systems development using a cyclical and iterative process generating multisectoral knowledge of locally relevant public health problems and associated determinants of health towards action. Throughout the process, the MSM-PRC fostered co-learning and capacity building, recognizing the strengths and resources within the community. Members of the CCB and DME co-created the CHNAA and were instrumentally involved in the decision-making of the scope and focus of the final survey. The community voice demonstrated through CHNAA survey responses helped the MSM-PRC to develop meaningful research questions in response to the local context of health disparities and their underlying determinants.

The CHNAA-driven interventions applied an ecological approach to focus on locally relevant public health concerns. In so doing, the CHNAA collected relevant community public health concerns specific to the communities served. For example, the Georgia Department of Public Health, through its hospitalization registry, report the 5 top health concerns, per 100,000 population, in the state are (1) Septicemia (604.4), (2) Hypertension and Heart Disease (360.9), (3) Mental Health Disorders (381.3), (4) Disease of Musculoskeletal and Connective Tissue (270.3), and (5) Ischemic Heart and Vascular Disease (261.5) [33] while the CHNAA revealed that diabetes and COVID-19 were of higher concern. However, the CHNAA noted that the top 5 health concerns reported by community members were diabetes, COVID-19, mental health, high blood pressure, and heart disease. Suggesting that surveillance and disease registry may not collect valid data for local concerns. In collaboration with the CCB, MSM-PRC disseminated key study findings through publications to promote long-term and sustainable changes in the community [3,10,14,21,23,29,34]. Through the facilitation of civic engagement, community concerns can then evolve to the creation and establishment of policies and procedures to promote and advance health equity that places community needs and empowerment at the forefront.

Additionally, the CHNAA process allowed the MSM-PRC to continue fostering and expanding its relationships with community members throughout the state. Evidence suggests that capacity building at the community level increases confidence, resulting in community-led or co-led health interventions that are more likely to be successful in creating positive health behavior changes [6,7,9,10,14,15,34,35,36,37,38]. The CHNAA acts as a catalyst for guiding the current future research initiatives of the MSM-PRC. It also acts as a tool for creating a community-driven approach to addressing social determinants of health within underserved communities. The CHNAA generates and integrates knowledge for designing and implementing community-based public health research interventions that promote mutual benefits for all partners—community and academic.

Application: A key feature of CBPR is the emphasis on community participation in the codesign of public health interventions through community coalition or advisory boards, research committees, and consultation. With oversight and guidance from the CCB, the results of 2021–2022 CHNAA were used to inform the MSM-PRC public health intervention and research agenda. MSM-PRC and community partners were funded to implement five extramurally funded projects Table 1 in 2023 and 2024 to address diabetes and COVID-19 disparities and related risk factors, totaling over $10 million over a 4–5-year period. These initiatives, aligned with the CHNAA findings: offer MSM-PRC four Mental Health First Aid trainings [38] to over 100 community members and professionals in response to the priority or concern related to mental and behavioral health; have expanded its research focus on social determinants of health through community-level policy and advocacy trainings in urban and rural counties in Georgia; and has expanded its portfolio of community-investigators and community principle investigators (REACH and GA RESTORES; Table 1). The CHNAA has been widely used as the baseline for interventions throughout the MSM-PRC [8,21,22,23,26,27,35]. All projects are CCB-informed, -engaged, -led or -co-led. Furthermore, community members (CCB or others) are often co-authors of research products that are generated from these public health interventions. This article is a clear example of this feature of CBPR, which offers an opportunity for bridging the research-practice gap.

## 5. Challenges and Limitations

In research, there are often limitations associated with data collection and interpretation of survey responses. Because this iteration of the CHNAA was the first time that the survey was disseminated throughout the state, it was necessary to make use of a convenience sample of individuals who attended community events and those who had electronic/online access to the survey. For the CHNAA, self-reporting was the data collection method used for capturing survey participants’ attitudes, beliefs, and experiences about their health and that within the community. Self-reporting through a valuable data collection tool can also introduce potential biases, such as recall and social desirability biases. For the CHNAA, self-reporting bias was minimized by ensuring that the survey questions were clear, concise, and easily understood by participants. The survey captured real-time health concerns and challenges, therefore recall bias did not affect the validity of the results. The anonymity of the participants and minimal interactions between researchers and community members during data collection also limits social disability bias, which could have led to distortions in survey findings. While being able to obtain a robust sample size (N = 750), counties in metro Atlanta, specifically Fulton, were overrepresented. This may have increased the over-representation of low-income African American participants which was a population group of interest being more likely to experience adverse health outcomes and reside in health resource desserts. Nonetheless, obtaining community participation in more rural counties, especially the southern parts of Georgia, though proven to be much more challenging, would have helped in diversifying the pool of survey participants. Future directions highlight the need for strategic and intentional recruitment methods to be developed outside of the Atlanta metro by continuing to increase MSM-PRC presence and strengthening partnerships in these underserved and hard-to-reach areas.

## 6. Conclusions

The CBPR approach has been declared, applied, and propagated as an essential framework for conducting research throughout MSM to achieve its vision to lead the creation and advancement of health equity [13,14,21,22,27]. Most explicitly, CPBR has been increasingly promoted through the Morehouse Model for Community Engagement, a blueprint for advancing health equity by applying social action and social justice constructs to research [35].

The CBPR-driven CHNAA process and community-engaged tool contain significant implications for public health practice. Previous research highlights that communities can serve as both leaders and participants in research, health education, and promotion efforts. The senior leadership of the CCB has provided strong guidance in leading CHNAA initiatives by adopting a collaborative model in which community members are and remain active participants in the research through implementation of the CHNAA process. It is important to understand that the PRC’s relationship with the community continues to build and grow a notable presence throughout Georgia over time. While community-academic partnerships are highly valuable and beneficial to research, they require strong and unwavering commitment, hard work, much attention to detail, and a sincere passion for promoting health equity along with a sincere willingness to listen and respond to health concerns and challenges of the community. Each community-academic partnership is unique within itself and is full of its own complexities and nuances and should be characterized by shared vision and mutual benefits as stipulated through the CBPR principles. Consequently, when the community-engaged research process is built on the findings of evidence-based or practice-based assessment tools such as the CHNAA, the process can result in more collaborative and equitable partnerships in all phases of research. With collaboration and intentionality in keeping the community at the center, the MSM-PRC has designed an evidence-based research agenda intended to advance health equity.

Furthermore, MSM’s method for community engagement has been utilized throughout the country, by various institutions, and is aptly named the Morehouse Model [35]. Health needs and assets assessments, through the involvement and active participation of the community, can function as a key facilitator of moving beyond research and data collection, to meaningful practice that results in positive and equitable health outcomes for all.

In conclusion, this paper reinforces the transformative potential of the CBPR and Continuum of CEnR frameworks in advancing health equity. By centering community voices and fostering genuine partnerships, researchers and practitioners can move beyond data collection to catalyze social change and improve health outcomes for all.

## 7. Future Directions

CBPR-driven CHNAA has the potential to strengthen community-engaged research. Future directions for CHNAA, that is adaptable beyond the MSM-PRC, should include expanding its reach through education and outreach to young adults and student populations. This survey dissemination and translation method can be achieved by developing targeted educational campaigns in high schools, colleges, and universities, partnering with student organizations and health science programs, and integrating CHNAA findings into academic curricula. Utilizing social media, podcasts, and digital tools will further enhance engagement, while internships and volunteer opportunities will encourage student participation in community-based research. Additionally, providing formal data training for CCB members and community partners will support CHNAA replication for local impact. Structured training programs on data collection, analysis, and interpretation will enhance the data literacy of CCB members. Establishing mentorship programs, certification opportunities, and peer-to-peer learning sessions will further strengthen community capacity. Another key area of focus is expanding rural community-academic research engagements through strengthening partnerships with local health departments and public health programs in colleges and universities within rural SVI areas. Collaborating with rural health and academic institutions to integrate CHNAA methods into research and community health programs, co-developing strategies with health departments, and securing joint funding for sustainable projects will ensure an ongoing positive health impact. Additionally, implementing train-the-trainer programs and organizing annual rural health summits will empower local health officials and educators to conduct independent assessments and interventions, ensuring long-term health benefits for underserved communities.

## Figures and Tables

**Figure 1 ijerph-22-01030-f001:**
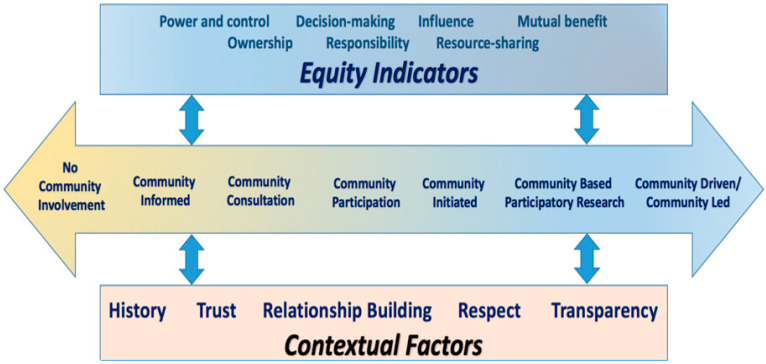
Continuum of Community Engagement in Research. Source: Reprinted from [9].

**Figure 2 ijerph-22-01030-f002:**
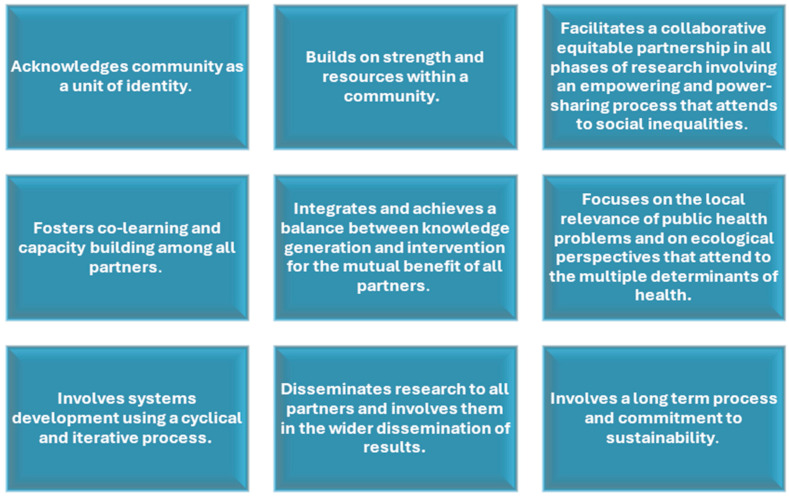
Community-Based Participatory Research Core Principles. Source: Adapted from [10,19].

**Figure 3 ijerph-22-01030-f003:**
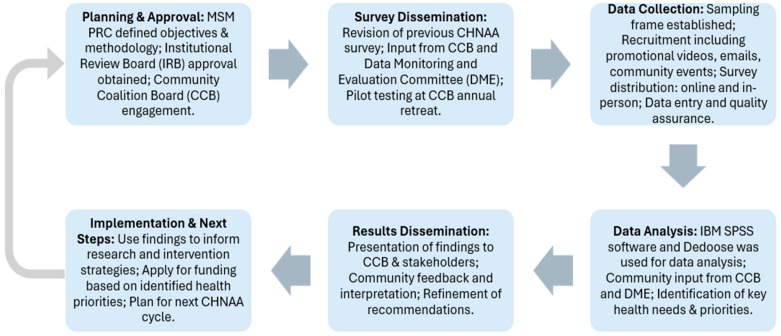
CHNAA Flow Chart.

**Table 1 ijerph-22-01030-t001:** CHNAA-Informed MSM-PRC Research Projects, 2023–2024.

Project Name	Funding Time Frame	Project Description	Population of Interest	Geographic Location
Mental Health First-Aid Training *	2024–2029	Training community members to learn, identify and manage stress and mental health.	Community members and professionals	Statewide
REACH **	2023–2028	Component A: Addresses risk factors for chronic diseases through nutrition, family weight management and physical activity.Component B: Adult vaccination—increase demand for COVID-19 and Flu vaccines through partnership and community-based collaboration	African American and Hispanic Adults	Fulton
GA CEAL RESTORES **	2024–2029	Family-based Diabetes prevention—family-based education to address pre-diabetes. Include mental health and civic engagement and policy advocacy training at the community level to increase support for families and address the social determinants of chronic diseases	African American and Hispanic pre-diabetic (Type 2) adults	Fulton, DeKalb, Hall and Macon Counties
MSM-PRC Core Research **	2024–2029	Family-based Diabetes Self-Management—Family-based intervention to address challenges with disease management through practical intervention (e.g., diet and nutrition) to prevent the exacerbation of reduce risk factors through connection with and support of community-based organizations.	African American and Hispanic Type 2 diabetic adults	Atlanta Metropolitan Area and Dalton City
PEACH 2 *	2023–2024	To evaluate a community-based, adaptive home-based COVID-19 testing program with behavioral nudges delivered via mobile phone texts to increase uptake of COVID-19 testing and prevention behaviors among individuals affected by diabetes (with diabetes, at risk for diabetes, or caring for someone with diabetes).	African American and Hispanic	Statewide

Source: https://www.msm.edu/Research/research_centersandinstitutes/PRC/preventionPrograms.php accessed on 23 March 2025. * CCB led; ** CCB co-led.

## Data Availability

Data is not publicly available. It is stored in secured; password protected server within MSM-PRC.

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
