# Peer review of "A Community-Engaged Approach to Community Health Needs and Assets Assessment for Public Health Research"

_ijerph, 2025, doi:10.3390/ijerph22071030_

Round 1
Reviewer 1 Report
Comments and Suggestions for Authors
Thank you for the opportunity to review this manuscript. Given that community-based participatory research is identified as the method, I am interested in learning more about how the Community Coalition Board was engaged in the various participatory processes e.g. were they involved in co-creating the questions, were they involved in analysis or interpretation or conclusions/recommendations? They were not acknowledged or listed as authors on this manuscript.
In addition, I am trying to understand how the research team did not require at least an Exemption Letter from the local IRB to undertake this research and publish this manuscript given that the manuscript was written from a research perspective. In addition, a statement regarding ethics is usually included in the last section of Methods.
As a reviewer, this work has great potential but without having the Community Coalition Board (tokenism) engaged in the various research processes, it would seem that the researchers would be less likely to be able to make sustainable changes. The gift of community-based participatory research is that it facilitates the building of relationships that overtime transform outcomes e.g. response rates are frequently higher, and change is sustainable at the local level.
Author Response
Thank you for your review. Please see attached for our responses.

Reviewer 2 Report
Comments and Suggestions for Authors
This manuscript describes efforts to include community voices in the CHNA process, with a detailed description of the survey used and the analysis of methods. It further describes some outcomes of the CHNA process, which if valid claims, are very important. However, the CHNA process is larger than what is described here, and other authors have made similar efforts at community engagement, which are not presented as context for this work. Below is a list of specific comments:
line 24-25: incomplete sentence
line 102-103 use continuum twice-not needed
what about the use of the CHNA process and products for non profit hospitals to meet IRS obligations-and thus identify their potential roles in community health improvement? Would help to describe the overall process, of which the survey described is just a part.
line 174-176 revise sentence;
line 188 Units plural?
line 213 CHNAA survey
p. 315 how does the CCB survey fit into the overall methodology? not mentioned as part of methods and data collection earlier
dissemination throughout publications? what about results directly back to communities themselves through events, discussions, community presentations? following through on CeNR process?
CHNAA process for research only? or to guide public health interventions?
There are other entities that have used CBPR methods in their CHNA; it would have been useful to include a summary of such work and how this work adds to the field.
CHNA is a larger process than just surveying community members. It would be helpful to describe the overall intent and purpose and make clear that what is being presented here is just a part of the overall process involving surveillance data, community partners, etc.
Author Response

(The authors gave the same response as above.)

Round 2
Reviewer 2 Report
Comments and Suggestions for Authors
In the Introduction, the statement The practice of not involving the community to address local
issues can erode trust, reduce cultural relevance, and limit the effectiveness of resulting interventions (lines 33-34) and the opening paragraph are concerning. The reference for public health interventions being based on surveillance systems is 15 years old-public health departments across the nation do some form of a MAPP process or other methods that include community involvement. I urge you to revise this very dated paragraph.
line 61-what are prioritized communities? prioritized by whom?
line 93 change has to have
line 95-96 reference for this? not sure it is true, if you use it, need to support
line 95 delete d from involved
line 112 so the purpose is a research agenda, not necessarily public health practice priorities; please be consistent with this-you state research priorities as purpose but then discuss practice priorities as outcomes
line 115 revise, not a grammatical sentence
line 122 consider changing this line to Community oversight is critical to protect its values, concerns, and interests. to Meaningful community engagement is critical to protect its values, needs and concerns.
line 148-149 revise sentence; not revising face validity but rather assuring it? also not revising comprehension but rather assessing it?
line 170 change in sampling is not As a Result of previous statement. It is a difference in methods
line 249 redundant; already gave gender breakdown in earlier sentence
line 272 phrase should NOT be italicized
line 287 on the CCB engagement assessment-were respondents asked which was most represented (one answer only)? If they were to answer all that applied, these frequencies are low overall. Please add description of what the question was.
line 308 what is a powered community?
line 376 last word should be though
line 385-386 if true is impressive, need to support with references
Of 36 references, 6 have no date and 6 are from before 2015. It would enhance the paper to have support from more current studies involving community engagement methods for CHNA
Comments on the Quality of English Language
This paper needs some editing. It was difficult to review due to grammatical errors. Also, 20% of references are old and updated versions are available.
Author Response
Thank you for your review and suggestions to enhance our work. Please see the attachment for Point-by-Point responses.
